# Perceived Stressful Life Events and Suicide Risk in Adolescence: The Mediating Role of Perceived Family Functioning

**DOI:** 10.3390/bs14010035

**Published:** 2024-01-03

**Authors:** Irene Caro-Cañizares, Nuria Sánchez-Colorado, Enrique Baca-García, Juan J. Carballo

**Affiliations:** 1Department of Psychology, School of Health and Educational Sciences, UDIMA (Universidad a Distancia de Madrid), 28400 Collado Villalba, Spain; 2Department of Psychiatry, Hospital Universitario Fundación Jiménez Díaz, 28015 Madrid, Spain; ebaca@quironsalud.es; 3Department of Psychiatry, Hospital Universitario Infanta Elena, 28342 Valdemoro, Spain; 4Department of Psychiatry, Hospital Universitario General de Villalba, 28400 Collado Villalba, Spain; 5Department of Psychiatry, Hospital Universitario Rey Juan Carlos, 28933 Móstoles, Spain; 6Department of Psychology, School of Health Sciences, Universidad Católica del Maule, Talca 3605, Chile; 7CIBERSAM (Centro de Investigación Biomédica en Red-Salud Mental), 28029 Madrid, Spain; juanjocarballo@yahoo.es; 8Department of Psychiatry, Centre Hospitalier Universitaire de Nîmes, 30900 Nîmes, France; 9Department of Psychiatry, Universidad Autónoma de Madrid, 28049 Madrid, Spain; 10Department of Child and Adolescent Psychiatry, Institute of Psychiatry and Mental Health, Hospital General Universitario Gregorio Marañón, 28007 Madrid, Spain; 11Instituto de Investigación Sanitaria Gregorio Marañón (IISGM), 28009 Madrid, Spain

**Keywords:** suicidal behavior, adolescence, stressful life events, family

## Abstract

Aim: Suicidal behavior is a serious public health problem and a major cause of death among adolescents. Three categories of major risk factors have been identified: psychological factors, stressful life events, and personality traits. Severe and objective stressful life events (SLEs), such as childhood mistreatment or abuse, have been clearly associated with higher rates of suicide risk. However, the relationship between suicide risk and adolescents’ perceptions of the SLE impact is not as clear. This paper studies the relationship between SLE impact perception and suicide risk and the possible mediating role of perceived family functioning in this relationship. The need for longer-term or more intense psychological or psychiatric treatment in relation to SLE impact perception is also addressed. Method: One hundred forty-seven adolescents aged 11–17 were consecutively recruited from the Child and Adolescent Mental Health Outpatient Services Department of a general hospital in Madrid, Spain. Self-informed questionnaires were used to assess suicide risk, SLEs, and family functioning. In addition, the clinical records of the participants were consulted to collect information about their treatment histories, including the number of appointments and the duration of follow-up. Results: SLE impact perception correlates significantly with suicide risk, the number of clinical appointments, the duration of treatment, and the perceived level of family functioning. The mediation model of the family functioning perception variable in the relationship between SLE impact perception and suicide risk is significant. The linear regression model of SLE impact perception and family functioning perception on suicide risk is also significant, accounting for 25.7% of the variance. Conclusions: Beyond the clear and proven effect of serious and objective SLEs, the perceived impact of SLEs reported by adolescents is related to an increased risk of suicide and more intense psychological and/or psychiatric follow-up. This relationship is mediated by the perceived level of family functioning. Adolescents’ perceptions of their life experiences and perceived family support may be key determinants of suicide risk prevention.

## 1. Introduction

Suicidal behavior is a challenge in the field of public health, and responses are often hindered by stigma and reluctance to address the issue [1]. The World Health Organization (WHO) warns that every year, approximately 700,000 people take their own lives intentionally [2]. Suicide is currently the second most common cause of mortality among people aged 15 to 24 [3]. The WHO expands this age group, indicating that in many countries, it is the second most common cause of death among young people aged 10 to 24 [4]. Added to these alarming numbers, we must pay attention to the rest of the manifestations of suicidal behavior. With regard to suicidal behavior, it is important to establish a stepwise differentiation, ranging from suicidal ideation at the mildest level, followed by suicide threats, attempts, and, ultimately, death by suicide, which represents the most serious manifestation [5,6]. Likewise, suicidal ideation also manifests as a spectrum that ranges from vague thoughts about not wanting to live to specific considerations about suicide. In this frame, death by suicide only represents 10% of all annual suicide attempts [7]. While it is true that most suicide attempts do not result in death, they are associated with other risks, such as serious injury, pain and suffering, and an increased likelihood of future suicide attempts [8,9]. Suicidal behaviors can also magnify adolescents’ broader experiences of mental health issues, which can cause a high degree of suffering and dysfunction in several crucial aspects of their lives [10].

Research carried out with adolescents from various cultures and backgrounds has shown that SLEs, such as physical or sexual abuse, family conflicts, and lack of social support from peers, are among the most significant risk factors associated with all forms of suicidal behaviors in all genders (e.g., [11]). Although they are rarely a sufficient cause for suicide or suicide attempts in isolation [12], research in this regard points out how the experience of a previous SLE explains around 10–15% of the variance in suicidal behavior [13]. Several studies have established a clear link between experiences of abuse in childhood and an increased risk of exhibiting self-harming behaviors in the future [14,15]. Psychological trauma can play a significant role in the predisposition to such behavior [16,17]. However, it is not only the objective presence of SLEs that can have consequences for adolescents’ mental health but also their perception of those experiences. It has been documented in adults, for example, that during a depressive episode, the subjective perception of stressful life events may be more related to suicidal behaviors than the objective number of such events [18]. On the other hand, adolescence is a period in which different daily stressors can become SLEs, which may have a higher likelihood of leading to self-harming behaviors. Factors contributing to higher stress levels may include poor academic performance or dropping out of school [19,20,21], contact with the criminal justice system [22,23,24], or acute stress and low overall quality of life [25,26].

Current research on suicidal behaviors focuses not only on identifying risk factors but also on recognizing protective factors. Social support and the presence of competent parents or caregivers are especially important [27,28,29]. Parental affection can act as a protective factor when adolescents face stressful situations, decreasing their likelihood of developing mental health problems [30]. Problems in family communication, a lack of solid emotional connections, excessive parental control, unstable family structures, a family history of suicidal behavior, and witnessing or experiencing domestic violence can restrict an adolescent’s ability to participate fully in their social environment and can complicate their ability to satisfy their most fundamental needs [31,32,33,34]. Similarly, it has been observed that neglect can increase the risk of suicide attempts among young people [35]. For these reasons, good family relationships are considered an important protective factor against the negative impact of life stressors [36].

Although research has clearly identified risk and protective factors, we are still not able to predict when suicidal behavior will take place [37]. This may be in part because adolescents’ impact perception of the SLEs they experience is not taken into account alongside the SLEs themselves. To the best of our knowledge, hardly any papers have been published on the possible mediating role of the perception of family functioning in the relationship between adolescents’ perception of the impact of SLEs and their risk of attempting suicide. The relationship between SLE impact perception and the intensity or duration of psychological treatment received has not been studied either.

Thus, the aims of this paper are twofold: first, to understand the relationship between SLE impact perception and suicide risk in a clinical sample of Spanish adolescents and determine whether their perception of family functioning acts as a mediating factor in this relationship, and second, to assess whether the intensity or duration of psychological or psychiatric treatment in adolescents is related to SLE impact perception.

Based on the previous literature, we hypothesize that SLE impact perception will correlate strongly with suicide risk and that this relationship will be mediated by family functioning perception. We will also explore the relationship between SLE impact perception and the intensity and duration of the psychological or psychiatric treatment.

## 2. Methods

### 2.1. Sample

As part of a larger research project, from September 2011 to October 2012, 267 patients undergoing initial evaluation at the Child and Adolescent Mental Health Outpatient Services of the Jimenez Diaz Foundation (Madrid, Spain) were consecutively recruited. The inclusion criteria included the patient’s age (from 11 to 17, both included) and the patient’s and parents’ ability to comprehend the questionnaires used. Patients who did not complete all of the questionnaires required were excluded. As previously published [38], analyses comparing the excluded and included patients found no differences in primary psychosocial characteristics. For the present study, all those who had turned 18 by 2016 were selected. The final sample consisted of 147 subjects. On 31 December 2016, all participants’ clinical histories were checked to collect follow-up data. The recruitment procedure and the characteristics of the sample have been previously published [39].

The study was approved by the Ethics Committee of the Jiménez Díaz Foundation (Madrid, Spain). Written informed consent was obtained from patients and parents or legal guardians.

### 2.2. Instruments

All participants were assessed on a clinical basis by experienced psychiatrists and completed the study questionnaires. Patients were administered the Spanish version of the Self-Injurious Thoughts and Behaviors Interview (SITBI) [40,41], a structured interview that assesses the presence, frequency, and characteristics of suicidal ideation; suicidal plans; suicidal gestures; suicide attempts; non-suicidal self-injury (NSSI) thoughts; and NSSI behaviors. A previous assessment of the Spanish version of the SITBI has proven that it has good psychometric properties, including interexam reliability (k > 0.09), test–retest reliability (Kappa index from 0.91 for suicidal ideation to 0.87 for suicide attempts), and construct validity (k = 0.99 for suicidal ideation and suicide attempts) [40].

The Spanish version of the Stressful Life Events Scale [42] was applied to obtain information regarding life stressors. This questionnaire asks respondents whether they have, in the past three years, experienced any of a list of 29 possible negative life events. Each item is scored as 1 if the event has occurred and 0 if it has not. For each event that is answered as having happened, participants rate, on a scale of 0 to 10, the degree to which it impacted them.

The third questionnaire administered, the Spanish version of the family APGAR (adaptability, partnership, growth, affection, and resolve) tool [43], is a five-item questionnaire that establishes perceived family functioning by asking the interviewee to rate, on a 3-point Likert scale (from 0, “almost never”, to 2, “almost always”), a series of key indicators. The total score allows a classification ranging from “functional family” to “severely dysfunctional family”. In its application with a Spanish sample [44], Cronbach’s alpha was 0.84.

Demographic data were obtained using a semi-structured interview developed ad hoc. Data related to age, gender, ethnicity, cohabitation status, socioeconomic level, and academic performance were collected.

Data on psychological and psychiatric follow-up, collected on 31 December 2016, included the total number of clinical appointments to date, the length of care in days, and, for those subjects who had reached the age of 18, whether or not they had been transferred to Adult Mental Health Outpatient Services.

### 2.3. Data Analysis

Suicide risk was assessed based on the sum of each participant’s responses to the suicidal ideation and suicide attempt subscales of the SITBI. In addition, SLE impact perception was assessed based on the sum of the degrees (from 0 to 10) to which the event has impacted each participant.

Pearson and point-biserial correlation coefficients were used to assess the strength of relations between variables considered in the study. In a second step, Bonferroni correction for multiple comparisons was applied to mitigate the risk of Type I errors when interpreting correlation coefficients. Linear regression analyses were conducted to examine the relationship between SLE impact perception, family functioning, and suicide risk. In the initial analysis, age and sex were included as covariates, but the results pointed out that age was not significantly related to the model, so only sex was included as a covariate. To delineate the interplay between SLE impact perception, family functioning, and suicide risk, four separate linear models were developed, with one independent variable each, as explained below:

Model 1: Suicide risk~sex as a covariate;

Model 2: Suicide risk~sex as a covariate + SLE impact perception;

Model 3: Suicide risk~sex as a covariate + family functioning;

Model 4: Suicide risk~sex as a covariate + SLE impact perception + family functioning.

A pairwise comparison of these models was conducted, comparing model 1 with model 2, model 1 with model 3, model 2 with model 4, and model 3 with model 4.

Mediation models were developed to assess the role of family functioning in the relationship between SLE impact perception and suicide risk. We followed standard methods for testing these models [45], which required meeting four criteria [46]: (1) the independent variable must be correlated with the dependent variable; (2) the independent variable must be correlated with the potential mediator; (3) the potential mediator must be correlated with the dependent variable, controlling for the independent variable; and (4) once the three previous conditions are met, the correlation between the independent and the dependent variables must decrease significantly with the inclusion of the potential mediator in the model. The analysis of mediation models was performed using bootstrap sampling methods. Bootstrapping is a nonparametric approach to test hypotheses, estimate size effects, and construct confidence intervals without making any assumptions about the shape of the distribution (normality, for example, which is needed in classical parametric methods). It is obtained by taking a large number of samples, with replacement, of size N from the data (where N is the original sample size) [47]. We used the INDIRECT macro [47] open syntax for SPSS to apply the bootstrapping method in the analysis of the mediation model. Once the mediation model is developed, a formal test is needed in order to determine the presence of the mediation effect [48]. Usually, the Sobel test is used; however, due to some limitations described for the Sobel test, especially when applied in small samples [47], the formal test to determine the presence of the mediation effect was also conducted with the INDIRECT macro. The independent variable (SLE impact perception) and the potential mediator (family functioning) were examined as continuous measures. For the first model, no other covariates were included. The effects of age and sex were controlled as covariates in a second model. 

## 3. Results

### 3.1. Sample Features

The final sample consisted of 147 subjects (61.2% male, 38.8% female) aged between 11 and 17 years (*M* = 15.34, *SD* = 1.316). Most were Caucasian (n = 128; 87.1%), lived with their family of origin (n = 126; 85.7%), and lived in a family with over 2000 euros/month in income (n = 60; 56.6%). Key characteristics of the sample are reported in Table 1 and were previously published [39].

Regarding suicidal thoughts and behaviors, 25.9% of the sample reported the presence of suicidal ideation (n = 38), 6.8% reported suicide attempts (n = 10), and 29.9% (n = 44) reported non-suicidal self-injury behaviors at least once in his or her life.

### 3.2. Correlation and Regression Analysis

Correlations between variables of interest are reported in Table 2. As expected, the total presence of SLEs significantly correlates with suicide risk (0.234; *p* = 0.005) and negatively correlates with the perceived level of family functioning (−0.302; *p* < 0.001). In addition, SLE impact perception significantly correlates with suicide risk (0.320; *p* < 0.001) and negatively correlates with the perceived level of family functioning (−0.339; *p* < 0.001). Among the follow-up variables, the total number of clinical appointments over the four years of the study significantly correlates with SLE impact perception (0.169; *p* = 0.004). The length of follow-up in days negatively correlates with SLE impact perception (−0.272; *p* = 0.001). Being transferred to the adult mental health service does not correlate with SLE impact perception.

Given that 35 pairwise correlation analyses were conducted, we applied Bonferroni correction for multiple comparisons to mitigate the risk of Type I errors. After Bonferroni correction, the statistical significance was set at *p* ≤ 0.0014 (0.05/35). Thus, SLE impact perception maintained its significant correlation with suicide risk (0.320; *p* < 0.001) and its negative correlation with the perceived level of family functioning (−0.339; *p* < 0.001). Among the follow-up variables, only the length of follow-up in days maintained its negative correlation with SLE impact perception (−0.272; *p* = 0.001).

Regarding the four linear regression models (Table 3), all four were significant: Model 1 accounted for 14.8% of the variance; Model 2 accounted for 18.9% of the variance; Model 3 accounted for 22.7% of the variance; and Model 4 accounted for 25.7% of the variance. The pairwise comparisons revealed the following:Comparing Model 1 (Adjusted *R*-square = 0.14.2, *p* < 0.001) with Model 2 (Adjusted *R*-square = 0.178, *p* < 0.001) showed that SLE impact perception makes a significant contribution to the model.Comparing Model 1 (Adjusted *R*-square = 0.136, *p* < 0.001) with Model 3 (Adjusted *R*-square = 0.216, *p* < 0.001) showed that family functioning makes a significant contribution to the model.Comparing Model 2 (Adjusted *R*-square = 0.178, *p* < 0.001) with Model 4 (Adjusted *R*-square = 0.24, *p* < 0.001) showed that family functioning and SLE impact perception taken together make a significant contribution to the model in which SLE impact perception is taken alone.Comparing Model 3 (Adjusted *R*-square = 0.216, *p* < 0.001) with Model 4 (Adjusted *R*-square = 0.24, *p* < 0.001) showed that family functioning and SLE impact perception taken together make a significant contribution to the model in which family functioning is taken alone.

Consequently, Model 4, in which SLE impact perception and family functioning (alongside sex and age as covariates) account for 25.7% of the variance, was selected as the more adjusted model.

### 3.3. Mediation Analysis

We followed standard methods to develop the mediation model via bootstrapping, with our results meeting the four criteria described above (see Figure 1). We found that (1) SLE impact perception significantly correlates with suicide risk; (2) SLE impact perception significantly correlates with family functioning perception; (3) family functioning perception significantly correlates with suicide risk, and this relationship remains significant when controlling for SLE impact perception; and (4) the relationship between SLE impact perception and suicide risk decreases when controlling for the potential mediator (family functioning perception). The model accounted for 13% of the variance (see Table 4). Studying the indirect effect via bootstrapping supports a partial mediation model, as the indirect effect is significantly different from zero at *p* < 0.05 (0.0067; [CI: 0.0018 to 0.0150]).

When controlling for sex and age (Figure 2), the results obtained continued to meet the four criteria described above, and the study of indirect effects supported a complete mediation model (0.0049; [CI: 0.0012 to 0.0121]). This model accounted for 25% of the variance (see Table 5).

## 4. Discussion

The primary aim of this study was to assess the relationship between the impact perception of SLE and suicide risk in a clinical sample of Spanish adolescents and the mediating effect of perceived family functioning. Several follow-up variables were also analyzed to determine their potential influence on SLE impact perception.

In relation to the prevalence of suicidal behavior, as previously reported [39], we found higher rates of suicide attempts than authors studying other Spanish samples [49], which is probably explained by our study’s focus on a clinical sample and the well-established increased prevalence of suicidal behaviors among people who suffer from mental health disorders [50]. In any case, it is important to note that suicide rates have still not been carefully studied in adolescence [51], and little research has been conducted with clinical outpatient samples [52]. Thus, the prevalence may vary depending on a number of variables, such as the instruments used, how suicidal phenomena are described, or the type of sample. 

Our analysis confirms findings reported in the literature [11,27,28,53,54,55] that SLEs and family functioning are significantly correlated with suicide risk. The sum of SLEs is a risk factor for the emergence of suicidal behaviors, while perceived good family functioning is a protective factor. In addition, our results suggest not only that the presence of SLEs is associated with a higher suicide risk but also that higher levels of perceived impact of SLEs are related to a higher suicide risk. This finding is of particular relevance since it focuses on adolescents’ subjective experiences, which, as has been reported for adult samples [18], seem to have an explanatory role in suicidal behavior. Unfortunately, hardly any existing publications address the subjective perception of the impact of SLEs on adolescents. Recently, it has been reported [56] that the associations seen between childhood mistreatment and a poor course of emotional disorders were largely attributable to the subjective experience of maltreatment.

For this reason, it seems urgent for research on SLEs among adolescents to devote greater attention to their perceived impact. Knowing the level of impact perceived by adolescents regarding the different SLEs that they report in consultation can help clinicians assess the risk of suicidal behavior and establish clinical objectives to buffer the SLE impact. These findings are consistent with those of studies that highlight resilience in the face of adversity as a protective factor (e.g., [57]).

On the other hand, we found that perceptions of family functioning played a mediating role in the relationship between SLE impact perception and suicide risk. Our results support the hypothesis that adolescents with poor family functioning are more likely to exhibit suicidal behaviors when they experience SLEs with greater perceived impact. Previous studies [27,28,57] have pointed out the buffering role of the family when facing SLEs, as well as its protective role with respect to suicide risk. Our study reveals that the family also plays a protective role against the perceived impact of SLEs on suicide risk. Knowing the level of family functioning perceived by an adolescent can help the clinician identify intervention objectives directly related to suicide risk prevention. In this context, it is important to note that there are some interpersonal elements, such as the perception of social and emotional support and the development of a solid therapeutic relationship, that reduce the likelihood of adolescents exhibiting suicidal behavior [10]. Recent works with network analysis pointed out that loneliness was a central factor for depression networks and also the most contributing factor to suicide ideation in adolescents [9,58]. Consequently, reinforcing the support, trust, and empathy provided by the family, strengthening communication skills, and identifying positive models in teachers and community members may reduce adolescents’ suicide risk [59]. 

Identifying the variables that predict suicidal behavior is relevant not only to the design of prevention strategies but also to the design of interventions for adolescents who are currently in treatment to reinforce the therapeutic alliance and prevent dropout [60]. Our results identify SLE impact perception as a relevant factor in clinical follow-up trajectories. Higher rates of SLE impact perception correlate with greater numbers of clinical appointments, which means that patients with higher perceptions of SLE impact are more likely to receive more frequent treatment. However, SLE impact perception correlates negatively with the length of follow-up, which may indicate that those patients are also more likely to abandon intervention prematurely. It would be reasonable to conclude that those who have experienced a greater subjective impact of SLEs—especially if they are linked to interpersonal events— are more likely to place less trust in the therapeutic bond [60]. According to Bowlby’s attachment theory [61], adolescents who perceive SLEs in their lives to have a greater impact may abandon therapy sooner because they have less experience with secure attachments, which makes it difficult to create and maintain the therapeutic bond. Furthermore, if SLEs are related to family problems, a lack of parental support may influence the decision to abandon therapy. This gives rise to the paradox that those patients who need follow-up the most may be the ones who abandon it the earliest. Thus, taking into account the perceived impact of SLEs reported by patients in consultation may help identify a risk of premature abandonment and help prevent it.

Despite the findings of this study, it has several limitations that must be taken into account and may affect the generalization of the results. First, it is important to note that our sample is not representative of the general adolescent population, so these results cannot be generalized beyond the clinical population. In addition, although the majority of the participants were native Spanish speakers (born in Spain, Colombia, Dominican Republic, Ecuador, Peru, or Uruguay, n = 139; 94.5%), there was a small percentage of the sample that came from other countries. This could have altered their comprehension of the questionnaires, which were all Spanish versions.

It is also important to note that we have treated the perceived impact of SLEs as a single cumulative variable instead of categorizing them based on different types of experiences (family relationship problems, peer relationship problems, abuse, violence, etc.), which could lead to different results. However, the sample size and number of SLEs reported did not allow us to analyze specific types of SLEs in detail. Likewise, due to the sample size, the level of perceived family functioning has been treated only based on the global measurement of the APGAR family scale. This has limited us from knowing how each of the five dimensions of the family APGAR is related to suicide risk.

Finally, we must point out a theoretical limitation related to mediation analyses. This type of analysis is based on confirmatory studies [62], which implies that results that support the hypothesis do not necessarily mean that the hypothesis is true. Therefore, to be able to confirm that a significant mediation model actually implies mediation in the data, a theoretical basis for the mediation effect must be developed prior to the analysis. Otherwise, there would be no way to distinguish a genuine mediation from a spurious relationship. In our case, based on the literature (e.g., [36]), we posited a mediation relationship before we started our analysis. It is reasonable to think that the relationship between the perception of the impact of SLEs and suicide risk is not spurious, as it is theoretically supported. Furthermore, it should be noted that our results for the first mediation model point to a partial mediation model since the relationship between SLE impact perception and suicide risk decreases when controlling for the potential mediator but does not disappear entirely. This means that family functioning perception does not fully account for the relationship between the perceived impact of SLEs and suicide risk. However, in the second mediation model, when controlling for sex and age, the results supported a complete mediation. Our results open the door to further studies on the relationship between the perceived impact of SLEs and suicide risk. In this regard, treating the family functioning as a single global measure prevented us from knowing which specific component(s) of the family relationship plays a significant role in mediating the effect of SLE impact perception on suicide risk.

## 5. Conclusions

Despite its limitations, this is the first study, to the best of our knowledge, that directly addresses the subjective perception of SLE impact and its relationship with suicide risk in adolescents. Furthermore, it highlights that this perceived impact effect on suicide risk may be mediated by the level of family functioning perceived by the adolescent. Our results also shed light on variables that must be carefully addressed to improve engagement with follow-up treatment. In conclusion, it is essential for clinicians and researchers to focus on the subjective experience of adolescents, and this is especially important for the reduction of suicide risk. Further studies with larger samples, more specific measures, and an exhaustive follow-up are warranted.

## Figures and Tables

**Figure 1 behavsci-14-00035-f001:**
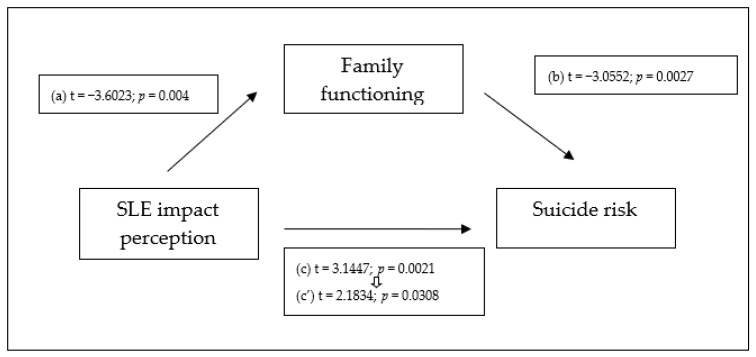
Family functioning perception partially mediates the relation between SLE impact perception and suicide risk. N = 135. (a) = Correlation between the independent variable (SLE impact perception) and the proposed mediator (family functioning perception); (b) = effect of the proposed mediator (family functioning perception) on the dependent variable (suicide risk), controlling for the independent variable; (c) = the total effect of the independent variable (SLE impact perception) on the dependent variable (suicide risk), not controlling for the mediator; (c’) = the effect of the independent variable (SLE impact perception) on the dependent variable (suicide risk), controlling for the proposed mediator (family functioning perception).

**Figure 2 behavsci-14-00035-f002:**
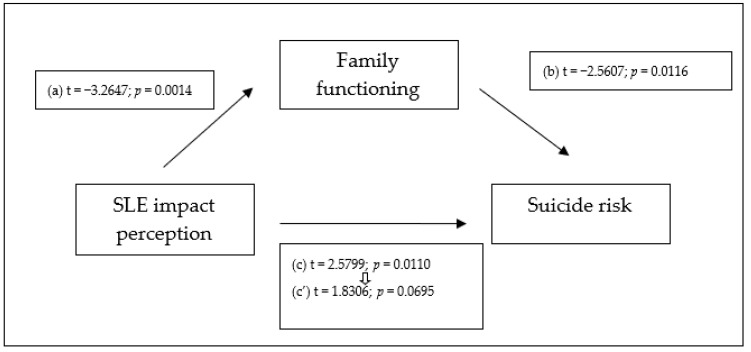
Family functioning perception completely mediates the relation between SLE impact perception and suicide risk when controlling for sex and age. N = 135. (a) = Correlation between the independent variable (SLE impact perception) and the proposed mediator (family functioning perception); (b) = effect of the proposed mediator (family functioning perception) on the dependent variable (suicide risk), controlling for the independent variable; (c) = the total effect of the independent variable (SLE impact perception) on the dependent variable (suicide risk), not controlling for the mediator; (c’) = the effect of the independent variable (SLE impact perception) on the dependent variable (suicide risk), controlling for the proposed mediator (family functioning perception).

**Table 1 behavsci-14-00035-t001:** Sociodemographic characteristics of the sample.

Total Sample	N = 147
	N (%), *M*, *SD*
Age (ranging from 11 to 17)	147 (100), 15.34, 1.316
	N (%), CI
Sex	147 (100)
Male	90 (61.2)
Female	57 (38.8)
Ethnicity	137 (93.2)
Caucasian	128 (87.1)
Latin American	1 (0.7)
Black	1 (0.7)
Gypsy	1 (0.7)
Others	6 (4.1)
Academic performance	143 (97.3)
Repeated course YES	65 (44.2)
Adopted	143 (97.3)
YES	13 (8.8)
Monthly income (EUR per capita)	106 (72.1)
>2500	33 (22.4)
2000–2500	27 (18.4)
1500–1999	18 (12.2)
500–1499	24 (16.3)
<500	4 (2.7)
Cohabitation status	146 (99.3)
Family of origin	126 (85.7)
Other relatives	2 (1.4)
Adoptive family	13 (8.8)
Institution	4 (2.7)
Other	1 (0.7)
Clinical diagnoses	146 (99.3)
Behavioral disorders	89 (60.5)
Emotional disorders	19 (12.9)
Anxiety disorders	16 (10.9)
Eating disorders	11 (7.5)
Other	7 (4.8)
No diagnosis	4 (2.7)

Note: *M*: median; *SD*: standard deviation; CI: confidence interval.

**Table 2 behavsci-14-00035-t002:** Pearson and point-biserial correlations, as applicable, between variables in the study.

	2	3	4	5	6	7	Sex	Age
	*R* (*p*)	*R* (*p*)	*R* (*p*)	*R* (*p*)	*R* (*p*)	*R* (*p*)	*R* (*p*)	*R* (*p*)
1. Suicide risk	0.321 (<0.001) **	0.234 (0.005) **	0.315 (<0.001) **	0.180 (0.031) *	−0.118 (0.158)	0.063 (0.453)	0.385 (<0.001) **	0.054 (0.519)
2. SLE impact perceived		0.681 (<0.001) **	−0.339 (<0.001) **	0.171 (0.039) *	−0.272 (<0.001) **	0.102 (0.219)	0.327 (<0.001) **	−0.056 (0.498)
3. SLE			−0.302 (<0.001) **	0.115 (0.167)	−0.264 (0.001) **	0.000 1	0.179 (0.03) *	−0.03 (0.714)
4. Family functioning				0.029 (0.736)	0.236 (0.006) **	0.122 (0.156)	−0.198 (0.020) *	0.184 (0.032) *
5. Number of clinical appointments					−0.107 (0.196)	0.443 (<0.001) **	0.182 (0.027) *	−0.010 (0.902)
6. Length of follow-up in days						0.104 (0.210)	−0.138 (0.095)	0.151 (0.068)
7. Transferred to adult mental health service							0.048 (0.546)	0.125 (0.131)

* *p* <0.05, ** *p* <0.001, *R*: correlation coefficient, *p*: probability.

**Table 3 behavsci-14-00035-t003:** Summary of regression models on suicide risk.

	Suicide Risk				
	F; df (*p*)	*R*	*R*-Square	Adjusted *R*-Square	Std. Error of the Estimate
Model 1	24,841; 1 (<0.001) **	0.385	0.148	0.142	0.555
Model 2	16,571; 2 (<0.001) **	0.435	0.189	0.178	0.544
Model 3	19,424; 2 (<0.001) **	0.477	0.227	0.216	0.528
Model 4	15,113; 3 (<0.001) **	0.507	0.257	0.24	0.520

** *p* < 0.001.

**Table 4 behavsci-14-00035-t004:** Mediation analysis.

IV to Mediators (a path)			
	Coeff	t	*p*
Family APGAR	−0.0948	−3.6023	0.0004
Direct Effects of Mediators on DV (b path)			
	Coeff	t	*p*
Family APGAR	−0.0709	−3.0552	0.0027
Total Effect of IV on DV (c path)			
	Coeff	t	*p*
SLE impact perception	0.0228	3.1447	0.0021
Direct Effect of IV on DV (c’ path)			
	Coeff	t	*p*
SLE impact perception	0.0161	2.1834	0.0308
Model Summary for DV Model			
*R*-square	Adjusted *R*-square	F	*p*
0.1307	0.1175	9.9214	0.0001
Bootstrap Results for Indirect Effects			
Indirect Effects of IV on DV through Proposed Mediators (ab paths)			
	Data	Boot	Bias
Total	0.0067	0.0069	0.0001
Family APGAR	0.0067	0.0069	0.0001
Bias-Corrected Confidence Intervals			
	Lower	Upper	
Total	0.0018	0.0150	
Family APGAR	0.0018	0.0150	

**Table 5 behavsci-14-00035-t005:** Mediation analysis controlling for sex and age.

IV to Mediators (a path)			
	Coeff	t	*p*
Family APGAR	−0.0842	−3.2647	0.0014
Direct Effects of Mediators on DV (b path)			
	Coeff	t	*p*
Family APGAR	−0.0579	−2.5607	0.0116
Total Effect of IV on DV (c path)			
	Coeff	t	*p*
SLE impact perception	0.0176	2.5799	0.0110
Direct Effect of IV on DV (c’ path)			
	Coeff	t	*p*
SLE impact perception	0.0127	1.8306	0.0695
Partial Effect of Control Variables on DV			
	Coeff	t	*p*
Sex	0.4098	4.2517	0.0000
Age	0.0326	0.9232	0.3576
Model Summary for DV Model			
*R*-square	Adjusted *R*-square	F	*p*
0.2523	0.2293	10.9646	0.0000
Bootstrap Results for Indirect Effects			
Indirect Effects of IV on DV through Proposed Mediators (ab paths)			
	Data	Boot	Bias
Total	0.0049	0.0048	−0.0001
Family APGAR	0.0049	0.0048	−0.0001
Bias-Corrected Confidence Intervals			
	Lower	Upper	
Total	0.0012	0.0121	
Family APGAR	0.0012	0.0121	

## Data Availability

Data supporting the reported results can be made available upon request.

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
