# Peer review of "Perceived Stressful Life Events and Suicide Risk in Adolescence: The Mediating Role of Perceived Family Functioning"

_behavsci, 2024, doi:10.3390/bs14010035_

Round 1

Reviewer 1 Report

Comments and Suggestions for Authors

Nice clean work, just few minor comments.

Good luck

Reviewer 2 Report

Comments and Suggestions for Authors

Dear Authors,

I read with interest your work, as suicide is an important theme that needs to be addressed.

I did not detect severe shortcomings in your manuscript: it is clear, coherent and methodologically adequate. I particularly appreciated your mediation analysis.

I have some suggestions to give you:

- Please correct "Crombach" (lines 141-142) in "Cronbach"; 

- p-values are referred to the probability of making a I-type error, so it is not recommended to write it with the symbol " = ". It is more correct to indicate it with the symbol " < ", as it defines an interval of probability. So I ask you to correct as follows: p = 0.005 line 182 --> p < .01, p = 0.004 line 187 --> p < .01; p = 0.001 line 188-189 -- > p < .01. Similar corrections should be done in the representation of mediation model (Figure 1); 

- In Table 3, "R square" should be changed in "R-squared". 

Focusing on your Introduction and Discussion parts, I would suggest you to consider recent works with Network Analysis, which could enrich your argumentations. I think this could be useful for you as that methodology is particularly useful to identify key factors for prevention. For instance, you can see this work on suicide: DOI 10.1002/cpp.2924 

I support the publication of your work after minor revision.

Reviewer 3 Report

Comments and Suggestions for Authors

Major comments:

1. The authors are advised to adjust for random incidents that may be mistakenly regarded as significant. For instance, in Table 2, where 35 pairwise correlation analyses are conducted, it would be prudent to apply a correction for multiple comparisons to mitigate the risk of Type I errors.

2. The rationale behind Table 3 is unclear. Are the authors attempting to quantify the percentage of variance in suicide risk attributable to stress life events (SLE) impact perception and family functioning? If that is the case, it is recommended that the authors pinpoint potential covariates, given the demographic data at their disposal. These covariates should be incorporated into the linear models. Furthermore, I would suggest delineating the interplay between SLE impact perception, family functioning, and suicide risk via four separate linear models with one independent variable each, as exemplified below using SLE impact perception:

Model1: Suicide risk ~ gender + other covariates

Model2: Suicide risk ~ gender + other covariates + SLE impact perception

Model3: Suicide risk ~ gender + other covariates + family functioning

Model4: Suicide risk ~ gender + other covariates + SLE impact perception + family functioning

A pairwise comparison of these models should be conducted, specifically comparing model 1 with model 2, model 1 with model 3, model 2 with model 4, and model 3 with model 4.

3. Mediation analysis and results.

(1) The depiction of the mediation analysis in the current Figure 1 is somewhat limited. I recommend that the authors: (1) consistently present the complete outcomes of the mediation analysis in a table, and (2) distinctly exhibit the results from model 2 and the mediation model (model 4) as separate sub-figures to avoid confusion.

(2) I wonder whether the authors have considered covariates in the mediation analysis.

(3) "Studying the indirect effect via bootstrapping supports the mediation model as the indirect effect is significantly different from zero at p< 0.05 (0.009, CI: 0.003 to 0.038)." When discussing the mediation results, such as the indirect effect's significance via bootstrapping, caution is warranted. The current Figure 1 suggests that the inclusion of the mediator variable does not fully account for the relationship between the independent and dependent variables, indicating only partial mediation. This interpretation should be clearly articulated.

(4) To enhance the interpretability of the results, the authors should elucidate the extent to which family functioning mediates the effect of SLE impact perception on suicide risk.

4. Given that the family APGAR questionnaire evaluates five dimensions—adaptability, partnership, growth, affection, and resolve—I am interested in learning which specific component(s) play a significant role in mediating the effect of SLE impact perception on suicide risk, including the mediation proportion.

Minor Comments:

1. It is advisable to utilize the most recent epidemiological data. The prevalence of suicide behaviors referenced on Lines 47-48 is outdated, dating back to 2014.

2. Regarding the questionnaires administered by psychiatrists, I am curious whether they were all in the Spanish version and if all the participants were native Spanish speakers.
